# Uncertainty Quantification and Decomposition for LLM-based Recommendation

Anonymous

## ABSTRACT

Despite the widespread adoption of large language models (LLMs) for recommendation, we demonstrate that LLMs often exhibit uncertainty in their recommendations. To ensure the trustworthy use of LLMs in generating recommendations, we emphasize the importance of assessing the reliability of recommendations generated by LLMs. We start by introducing a novel framework for estimating the predictive uncertainty to quantitatively measure the reliability of LLM-based recommendations. We further propose to decompose the predictive uncertainty into recommendation uncertainty and prompt uncertainty, enabling in-depth analyses of the primary source of uncertainty. Through extensive experiments, we (1) demonstrate predictive uncertainty effectively indicates the reliability of LLM-based recommendations, (2) investigate the origins of uncertainty with decomposed uncertainty measures, and (3) propose uncertainty-aware prompting for a lower predictive uncertainty and enhanced recommendation. Our source code and model weights are available at https://anonymous.4open.science/r/UNC_LLM_REC

## CCS CONCEPTS

• **Information systems** → **Collaborative filtering**; **Personalization**; **Recommender systems**.

## KEYWORDS

Recommendation; Large Language Models, Uncertainty

**ACM Reference Format:**

Anonymous Author(s). 2025. Uncertainty Quantification and Decomposition for LLM-based Recommendation. In *Proceedings of Proceedings of the ACM Web Conference 2025 (WWW '25)*. ACM, New York, NY, USA, 12 pages. https://doi.org/XXXXXXX.XXXXXXX

## 1 INTRODUCTION

Large language models (LLMs) have recently been widely adopted for recommendation [2, 19, 79], as they have powerful comprehension ability for context [23] and textual features [25]. LLMs, pre-trained on an enormous corpus, possess a wealth of external knowledge for open-domain tasks [4, 6, 70] (e.g., Iron Man and Spider-Man share the same universe, the red wine goes well with beef), and can be leveraged for recommendations that require background information or common sense. Instruction-tuned

LLMs [4, 29, 62, 64] have shown remarkable performance for the zero-shot ranking task [23, 25], and can be further fine-tuned with the user history logged on the system [2, 19, 79]. Recent methods [10, 68, 77, 78] adopt the retrieval-augmented generation paradigm [3, 27], where LLMs are employed to generate ranking lists with candidates retrieved by candidate generators. This approach exhibits state-of-the-art recommendation performance over conventional sequential recommenders [31, 61], facilitating better online updates and avoiding hallucination.

While LLMs have been widely employed in real-world applications that can influence human behavior, there is a lack of exploration in assessing the reliability of the LLM-based recommendation. Indeed, despite their superior performance, we demonstrate recommendations generated by LLMs are highly volatile depending on the prompting details (e.g., word choice, number of user histories, number of candidate items), despite using the same user history and candidates. Moreover, we observe the volatility in generation correlates with the recommendation performance. This newly-arisen challenge is specific to LLM-based recommendation, which does not occur with conventional recommender models [22, 38, 56]. Therefore, we assert the need to assess the reliability of recommendations generated by LLMs, to support the trustworthy use of the advanced comprehension capabilities of LLMs. Although significant progress has been made in estimating the reliability of LLMs primarily in classification [44, 63, 72, 74] and question-answering [39, 40], there is a lack of study focused on recommendation generated by LLMs.

Our work fills this gap by attempting to improve our understanding of the reliability of LLM-based recommendations. We start by quantitatively measuring the reliability of LLMs for recommendation, by embracing the concept of *predictive uncertainty* [12, 16, 34]. The predictive uncertainty can be understood as the entropy of predictive distribution [48], and has been utilized to indicate the reliability of responses generated by LLMs for various tasks [39, 40, 44]. If LLMs generate volatile recommendations across multiple inferences with given user history and candidates, the predictive distribution would be smooth and the predictive uncertainty is high. Conversely, if LLMs consistently produce identical recommendations across all generations, the predictive distribution would manifest as a one-hot vector and the predictive uncertainty is zero. Therefore, the estimated predictive uncertainty helps us gauge how much we can trust the recommendations made by LLMs.

However, assessing the predictive uncertainty of LLM-based recommendation raises a challenge as the output space of possible ranking lists is intractable, compared to more confined output spaces of classification [12, 20, 44] and multiple-choice question answering [39, 40]. The output ranking space has a size equal to the factorial of the number of retrieved candidates, and therefore, utilizing LLMs' autoregressive generation probability would be infeasible to cover the entire output space. To tackle this limitation, we estimate the ranking probability by adopting Plackett-Luce

model [47, 52] with the top-1 probability of candidate items. By doing so, we can approximate the predictive distribution for the entire ranking space with a *single* inference, and obtain the predictive uncertainty as the entropy of the estimated predictive distribution.

We further propose decomposing the predictive uncertainty to identify the primary source of uncertainty in LLM-based recommendations. Specifically, we introduce a latent variable representing the prompt, and decompose the total predictive uncertainty into *recommendation uncertainty* and *prompt uncertainty*. The recommendation uncertainty denotes the intrinsic uncertainty originating from the recommendation difficulty of the user history and candidates. On the other hand, the prompt uncertainty denotes the uncertainty raised from the prompting scheme, which is an additional volatility when recommendations are generated by LLMs. These decomposed uncertainty measures serve as a tool for in-depth analyses to gain a deeper understanding of the various factors influencing the performance of LLM-based recommendations.

Based on our framework for quantifying and decomposing the uncertainty of LLMs for recommendation, we conduct an extensive empirical investigation on real-world datasets. We offer our key findings and contributions throughout this paper as follows:

- **Predictive uncertainty indicates reliability of recommendation (Section 6).** We demonstrate the effectiveness of estimated predictive uncertainty; a recommendation with lower uncertainty yields higher recommendation performance. Moreover, our uncertainty quantification framework shows superiority to existing methods for other natural language tasks, with less or comparable inference burden.
- **Unveiling origins of uncertainty with uncertainty decomposition (Section 7).** We investigate the effect of various factors (e.g., fine-tuning, model size, user history, and candidate items) on the decomposed uncertainty measures, and analyze how it is related to recommendation performance. Our observations provide promising explanations for the previously raised LLM-specific limitations; a larger number of user histories and candidate items may bring increased uncertainty and thus do not necessarily improve recommendation performance.
- **Enhancing recommendation with uncertainty-aware prompting (Section 8).** Based on our insights above, we support the best use of LLMs for recommendation. We propose to adjust the number of user histories and candidate items, for a lower predictive uncertainty and further enhanced recommendation. Our uncertainty-aware prompting methods improve recommendation performance with negligible changes in the number of tokens used.

## 2 RELATED WORK

**Large Language Models for Recommendation.** Large language models (LLMs) have recently gained widespread adoption in recommendation systems due to their powerful ability to comprehend complex contexts and to utilize external knowledge for generation [11, 71]. Early work [9, 19, 61] adapts the architectures of language models for the recommendation task and outperforms conventional matrix factorization architectures [22, 38, 56]. As cutting-edge LLMs [29, 62, 64], pre-trained on extensive corpora and distributed publicly, show remarkable performance in open-domain tasks [4, 6, 70],

```
<<SYSTEM>>
You are a movie recommender system.
Given user's watch history, you output the index of recommended movie.
<<USER>>
I have watched the following movies in the past in order:
'{item}'    \\user history
'{item}'
...
The {num} candidate movies are as follows:
Candidate [A]: '{item}'    \\candidate items
Candidate [B]: '{item}'
...
Rank the candidate movies above based on their relevance to my watch history.
Only respond with the identifier of the movie without any word or explain.
```

**Figure 1: Example prompt for list-wise ranking with LLMs.**[1]

subsequent research highlights their effectiveness for the zero-shot [18, 23, 25, 68] and few-shot [45, 58] ranking task. Moreover, recent methods [2, 21, 33, 43, 79] involve fine-tuning LLMs with instruction on recommendation datasets, to mitigate the disparity between natural language understanding tasks used to train LLMs and the recommendation task [2, 35]. Lately, the state-of-the-art methods [10, 25, 68, 77, 78] adopt a list-wise ranking paradigm with the retrieval-augmented generation [3, 27]. In this approach, the generation of ranking lists is conditioned on candidates retrieved by candidate generators as illustrated in Figure 1, to better facilitate updates and reduce hallucination [11]. We refer readers to [11, 71] for a detailed survey.

**Uncertainty of Large Language Models.** In the era of LLMs, where human behaviors are influenced by the outputs of these models, recent research underscores the imperative of evaluating the reliability of LLM-generated responses [1, 72–74]. The predictive uncertainty [39, 44], quantified as the entropy of the predictive distribution, is widely adopted to measure the reliability of LLMs [12, 16, 34]. Despite significant progress in uncertainty estimation, with a primary focus on classification [44, 63, 72, 74] and question-answering [39, 40], there is a scarcity of research on the uncertainty of recommendations generated by LLMs. The main challenge of applying the aforementioned uncertainty quantification framework lies in the vast output space of possible ranking lists, compared to the more confined output spaces of classification [12, 20, 44] and multiple-choice question answering [39, 40]. Therefore, the existing methods requiring access to the entire predictive distribution cannot directly be applied to the uncertainty quantification of LLMs for recommendation, and a solution tailored to LLM-based recommendation is required.

**Uncertainty of Recommendation.** Uncertainty quantification remains an underexplored area within the recommendation literature. In the realm of recommendation systems, the uncertainty often refers to the unpredictable nature of user preferences [14, 30, 53, 66, 75] (e.g., users occasionally click whimsical items far from their preference). They introduce variance terms to model the variability inherent in user preferences, with the goal of enhancing recommendation performance using noisy interaction data. On the other hand, other works focus on the calibration of recommender models' output scores [20, 41], to select the threshold for the retrieval [7, 67] or to balance exploration-exploitation trade-off in multi-armed bandits [5, 60]. While a recent work [51] has investigated the variance of recommender models' output, this work is limited to simplistic binary classifiers for predicting click-through rates [80] and cannot

---

[1]It is worth noting that the goal of this paper is not to propose a high-performance prompting scheme; more sophisticated methods [76, 81] can be applied to our work.

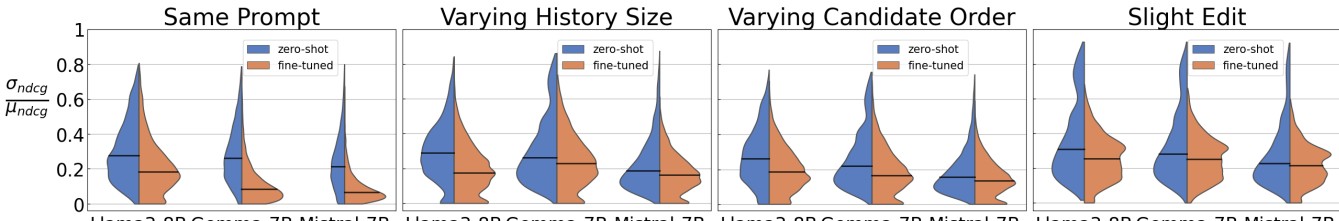

Figure 2: Violin plots describing the user distributions with the coefficient of variation.

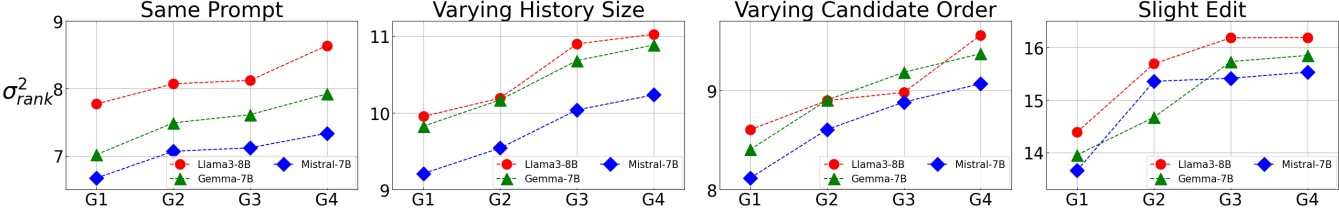

Figure 3: Generation volatility and recommendation performance of *fine-tuned* LLMs. We plot the rank variance of candidate items for each user group. G1 has the highest average N@20 and G4 has the lowest average N@20.

be extended to the LLMs for the list-wise ranking task. To the best of our knowledge, uncertainty quantification specifically tailored for LLM-based recommendation remains unaddressed. We believe our work fills this gap and has a broad impact, considering the transformative potential of LLMs in recommendation.

## 3 PRELIMINARIES

**Notations.** Let $\mathcal{U}$ and $\mathcal{I}$ denote a set of users and a set of items, respectively. For a user $u \in \mathcal{U}$, $\mathcal{H}_u = [i_{u,1}, i_{u,2}, \cdots, i_{u,|\mathcal{H}_u|}]$ denotes the sequential user history logged in chronological order of interaction time. Each item $i \in \mathcal{I}$ is associated with a representative text $\mathbf{t}_i \in \mathcal{T}$, which may serve as a title or description of the item.

**Two-Stage Recommendation.** Real-world recommender systems often employ a two-stage recommendation approach for efficient personalization [32, 67]. The first stage is *candidate generation* where heuristics or lightweight models are adopted for fast retrieval. They generate a small candidate set $C_u = \{c_{u,1}, c_{u,2}, \cdots, c_{u,|C_u|}\}$ from a tremendous number of items ($|C_u| \ll |\mathcal{I}|$). It is noted that classic candidate generation approaches do not assign any specific order for candidate items [8, 25]. In the subsequent stage, referred to as *ranking*, sophisticated models are developed to rank the candidate items by leveraging both the user history $\mathcal{H}_u$ and fine-grained features (e.g., $\mathcal{T}$) to derive the final ranking $\pi_u \in \Pi_u$. Here, $\Pi_u$ is the entire ranking space with a size of $|\Pi_u| = |C_u|!$.

**Large Language Models for List-wise Ranking.** Literature on LLMs for recommendation mainly focuses on the ranking stage, given that their inference is computationally intensive when applied to a large candidate set [25]. LLMs generate fine-grained ranking based on prompts constructed with the representative text of items in $\mathcal{H}_u$ and $C_u$. In this study, we employ a list-wise ranking approach [25, 76, 78] as illustrated in Figure 1. The list-wise ranking is more efficient than point-wise [2, 45, 79] and pair-wise [54] approaches, as these latter approaches require multiple prompts and model inferences to generate a single ranking list. The position of candidate items in the prompt is randomly assigned as candidate generation models produce unordered sets, as noted earlier

[8, 25]. Additionally, we assign an index to each candidate item and instruct LLMs to respond with these indices. The generated output for the list-wise prompt would be the complete ranking list of indices (e.g.,"C,A,B,..."). Otherwise, semantic parsing with text-matching algorithms [36] is required to compare the generated output with the representative texts of candidate items [25].

## 4 MOTIVATING ANALYSIS

We present our motivating analysis demonstrating that LLMs exhibit volatile recommendation performance even after fine-tuning. Furthermore, we show that this volatility in generation varies among users and correlates with the recommendation performance.

**Analysis setup.** We adopt three popular instruction-tuned LLMs, including Llama3-8B [64], Gemma-7B [62], and Mistral-7B [29] on MovieLens 1M dataset[2]. The candidates are retrieved with BPR-MF [56]. We set the default prompt as shown in Figure 1 and devise four additional prompt schemes with slight variations:

- **Same Prompt**: We use the exact same prompt for five stochastic[3] generations, utilizing 20 latest histories and 20 candidates.
- **Varying History Size**: We use the latest histories of different sizes from {10, 15, 20, 25, 30} for each deterministic[4] generation.
- **Varying Candidate Order**: We use shuffled candidate orders for each deterministic generation.
- **Slight Edit**: We apply minor format changes to the prompt (e.g., "Candidate [A]" → "Movie [A]" in Figure 1) for each deterministic generation.

For each user, we generated five recommendations $\{\pi_u^n\}_{n=1}^5$ for each prompt scheme. Please refer to Figure 5 in Appendix A for full prompts.

**M1: LLMs exhibit volatile recommendation performance for individual users.** To examine the volatility of LLMs in recommendation performance, we compute the mean and the standard

---

[2]https://grouplens.org/datasets/movielens/
[3]We sample the next token with predicted token probabilities and default temperatures.
[4]We select the next token with the highest token probability.

deviation of NDCG@20 [28] across the five generated recommendations $\{\pi_u^n\}_{n=1}^5$ for each user $u$. We then investigate the coefficient of variation [42] to understand the performance deviation relative to the average performance:

$$\frac{\sigma_{\text{ndcg}}}{\mu_{\text{ndcg}}} = \frac{\text{Var}_n[\text{ndcg}(\pi_u^n)]^{0.5}}{\text{E}_n[\text{ndcg}(\pi_u^n)]}, \tag{1}$$

where $\text{ndcg}(\pi_u^n)$ denotes NDCG@20 of ranking list $\pi_u^n$. Figure 2 shows the violin plots [24] describing the user distributions with the coefficient of variation.[5] The average coefficient of variation is around 0.3 for the zero-shot setup, which indicates the recommendation performance may fluctuate by 30%. The fine-tuned LLMs exhibit a lower coefficient of variation, but still, the recommendation performance fluctuates by 10-20% on average. This result suggests that even when the same prompt is used across five generations ('Same Prompt'), LLMs exhibit significant volatility in recommendation performance. Moreover, when LLMs generate recommendations in a deterministic manner (i.e., without any sampling), the volatility in recommendation performance increases with variations in prompting details (i.e., 'Varying History Size', 'Varying Candidate Order', 'Slight Edit').

**M2: Generation volatility correlates with the recommendation performance.** We divide users into four equally-sized groups based on their recommendation performance $\mu_{\text{ndcg}}$. For each user group $G$, we assess the variation in ranking orders across the five generated recommendations $\{\pi_u^n\}_{n=1}^5$:

$$\sigma_{\text{rank}}^2 = \text{E}_{u \in G}[\text{E}_{i \in C_u}[\text{Var}_n[\text{rank}(i; \pi_u^n)]]], \tag{2}$$

where $\text{rank}(i; \pi_u^n)$ denotes the rank of $i$ in $\pi_u^n$. Figure 3 illustrates the generation volatility (i.e., $\sigma_{\text{rank}}^2$) of fine-tuned LLMs for each user group. We observe that the user group with the highest recommendation performance (G1) demonstrates the lowest generation volatility in the rank of candidate items. In essence, LLMs consistently produce similar rankings regardless of stochastic generation or prompt variation when their predictions are relatively accurate.

**Measuring generation volatility to indicate the reliability of LLMs and enhance recommendation.** Despite the superiority of LLMs against conventional sequential recommenders [35, 79], our in-depth analyses demonstrate that LLMs exhibit large coefficients of variation in performance. Therefore, we highlight the importance of a comprehensive understanding of generation volatility to support the trustworthy use of the advanced capabilities of LLMs for recommendations. Throughout this paper, we present a systematic approach for (1) quantifying the generation volatility to indicate the reliability of LLM-based recommendations, (2) unveiling the origins of generation volatility, and (3) enhancing recommendations through prompting schemes that result in lower generation volatility.

## 5 UNCERTAINTY QUANTIFICATION AND DECOMPOSITION FOR RECOMMENDATION

### 5.1 Overview

We embrace the concept of *predictive uncertainty* [12, 16] to quantitatively measure the reliability of LLM-based recommendations.

To this end, we present a novel framework to estimate the predictive uncertainty of LLMs for recommendations, with the following contributions.

- **(Sec 5.2) Predictive Uncertainty Quantification**: We estimate the predictive distribution for recommendation with a single LLM inference and derive the predictive uncertainty from it.
- **(Sec 5.3) Uncertainty Decomposition**: We further decompose the predictive uncertainty into prompt uncertainty and recommendation uncertainty, to identify the origins of uncertainty.

The predictive uncertainty (1) serves as an indicator of reliability, and (2) can be further utilized to enhance recommendations with lower uncertainty.

### 5.2 Uncertainty Quantification

The (total) predictive uncertainty for recommendation is quantitatively measured by the entropy of $q(\pi_u|\mathcal{H}_u, C_u)$, the predictive distribution over potential recommendations $\pi_u$ for $\mathcal{H}_u$ and $C_u$:

$$\text{H}[q(\pi_u|\mathcal{H}_u, C_u)] = \text{E}_{q(\pi_u|\mathcal{H}_u, C_u)}[-\log q(\pi_u|\mathcal{H}_u, C_u)]. \tag{3}$$

If LLMs generate volatile recommendations with $\mathcal{H}_u$ and $C_u$, $q(\pi_u|\mathcal{H}_u, C_u)$ would be smooth and the predictive uncertainty is high. Conversely, if LLMs consistently produce identical recommendations across all generations, the predictive distribution $q(\pi_u|\mathcal{H}_u, C_u)$ would manifest as a one-hot vector and the predictive uncertainty is 0.

*5.2.1 **Defining predictive distribution**.* We introduce a latent variable $\mathcal{P}_u$ representing the prompt constructed with $\mathcal{H}_u$ and $C_u$. The prompting process $q(\mathcal{P}_u|\mathcal{H}_u, C_u)$ embraces all prompting details including system instruction, user messages, and orders of candidates. Then, we define the predictive distribution over potential prompts constructed with $\mathcal{H}_u$ and $C_u$ as follows:

$$q(\pi_u|\mathcal{H}_u, C_u) = \text{E}_{q(\mathcal{P}_u|\mathcal{H}_u, C_u)}[q(\pi_u|\mathcal{P}_u)]. \tag{4}$$

A naive approach for estimating $q(\pi_u|\mathcal{P}_u)$ is utilizing LLMs' autoregressive probability for sequentially generating $\pi_u$ with $\mathcal{P}_u$: $q(\pi_u|\mathcal{P}_u) = \prod_{k=1}^{|C_u|} q(\pi_{u,k}|\mathcal{P}_u, \pi_{u,1}, \cdots, \pi_{u,k-1})$. However, generating all ranking lists in $\Pi_u$ to obtain $q(\pi_u|\mathcal{P}_u)$ would be infeasible to cover the entire ranking space $\Pi_u$ with a size of $|C_u|!$ (e.g., $20! = 2.4 \cdot 10^{18}$ inferences when $|C_u| = 20$).

*5.2.2 **Estimating predictive distribution**.* As a workaround, we approximate the predictive distribution with the top-1 probabilities for $i \in C_u$ by adopting Plackett-Luce model [47, 52], which has been widely adopted to model ranking permutations [37]. First, to obtain top-1 probabilities for $i \in C_u$, we slightly modify our prompt in Figure 1 by replacing "Rank the candidate movies above" with "Which movie would I like to watch next most?". This prompting provides a format guideline and ensures that the LLMs follow the instruction [69, 82]. The generated output for this prompt would be an index (e.g.,"B") for an item in $C_u$. Then, with the output logit values $z_i \in \mathbb{R}$ for $i \in C_u$ (i.e., $q(i|\mathcal{P}_u) \propto \exp z_i$), we estimate the prompt-conditional generation probability for all possible ranking lists with a *single* inference:

$$\hat{q}(\pi_u|\mathcal{P}_u) = \prod_{k=1}^{|C_u|} \frac{\exp(z_{\pi_{u,k}})}{\sum_{l \in \mathcal{N}_k} \exp(z_l)} \approx \prod_{k=1}^{K} \frac{\exp(z_{\pi_{u,k}})}{\sum_{l \in \mathcal{N}_k} \exp(z_l)}, \tag{5}$$

---

[5]The average performance $\text{E}_{u \in \mathcal{U}}[\mu_{\text{ndcg}}]$ is reported in Table 4, Appendix B.

where $\pi_{u,k}$ is the $k$-th item in $\pi_u$ and $\mathcal{N}_k$ is the domain for the $k$-th sampling without replacement ($\mathcal{N}_k = \{\pi_{u,k}, \pi_{u,k+1}, \cdots, \pi_{u,|C_u|}\}$). Since multiplying $|C_u|$ probabilities results in *probability vanishing* problem, we multiply the probabilities only for top-$K$ candidates. Finally, we estimate the predictive distribution as follows:

$$\hat{q}(\pi_u|\mathcal{H}_u, C_u) = \mathbf{E}_{q(\mathcal{P}_u|\mathcal{H}_u, C_u)}[\hat{q}(\pi_u|\mathcal{P}_u)]. \tag{6}$$

In this work, we estimate the expectation by adopting Monte-Carlo method [57], sampling five prompts from $q(\mathcal{P}_u|\mathcal{H}_u, C_u)$.[6]

*5.2.3* ***Estimating predictive uncertainty****.* The next step is estimating total predictive uncertainty in Eq.3 with the predictive distribution estimated with Eq.6:

$$\mathbf{H}[q(\pi_u|\mathcal{H}_u, C_u)] \approx \mathbf{E}_{\hat{q}(\pi_u|\mathcal{H}_u, C_u)}[-\log \hat{q}(\pi_u|\mathcal{H}_u, C_u)]. \tag{7}$$

We draw ranking lists from $\Pi_u$ by simulation based on the sampling probability in Eq.5 and estimate the predictive uncertainty with Monte-Carlo method. It is worth noting that sampling without replacement can be readily done within a few ms with the exponential-sort trick [13].

## 5.3 Uncertainty Decomposition

The total predictive uncertainty encompasses uncertainties stemming from diverse sources. Therefore, investigating the primary source of uncertainty proves challenging. For instance, in recommendation, uncertainty might arise from the prompting scheme $\mathcal{P}_u$ or recommendation difficulty of $\mathcal{H}_u$ and $C_u$. Literature on Bayesian neural networks (BNNs) [12, 44] decomposes the total predictive uncertainty into aleatoric (data) uncertainty and epistemic (model) uncertainty, by employing an ensemble of multiple models. However, employing an ensemble of multiple cutting-edge LLMs is unavailable and inefficient in practice [39].

*5.3.1* ***Uncertainty decomposition through a latent variable****.* We decompose the total predictive uncertainty through a latent variable $\mathcal{P}_u$, without any burden from employing multiple LLMs. First, we measure the uncertainty arising from the prompting scheme by the mutual information between the prompt and the ranking:

$$\mathbf{I}[\mathcal{P}_u, \pi_u] \coloneqq \mathbf{E}_{q(\mathcal{P}_u, \pi_u)}\left[\log \frac{q(\mathcal{P}_u, \pi_u)}{q(\mathcal{P}_u)q(\pi_u)}\right]. \tag{8}$$

Here, we omit the condition on $\mathcal{H}_u$ and $C_u$ for brevity. If the conditional distribution $q(\pi_u|\mathcal{P}_u)$ remains the same regardless of prompt $\mathcal{P}_u$, the mutual information is 0. Conversely, when LLMs generate volatile recommendations depending on the prompting scheme, the mutual information would be high. Since the mutual information is formulated as $\mathbf{I}[X, Y] = \mathbf{H}[Y] - \mathbf{E}_{q(X)}[\mathbf{H}[Y|X]]$, the total predictive uncertainty can be decomposed as follows:

$$\underbrace{\mathbf{H}[q(\pi_u|\mathcal{H}_u, C_u)]}_{\text{Total Unc.}} = \underbrace{\mathbf{I}[\mathcal{P}_u, \pi_u]}_{\text{Prompt Unc.}} + \underbrace{\mathbf{E}_{q(\mathcal{P}_u|\mathcal{H}_u, C_u)}[\mathbf{H}[q(\pi_u|\mathcal{P}_u)]]}_{\text{Recommendation Unc.}}. \tag{9}$$

We denote the first term on the right-hand side as *prompt uncertainty*, measuring the uncertainty raised by the prompting scheme. This uncertainty is specific to LLM-based recommendation and does not occur with conventional recommender models [22, 38, 56].

---

[6]It is noted that we can sample just one prompt when only obtaining the total uncertainty without uncertainty decomposition.

---

**Algorithm 1:** Uncertainty in LLM-based Recommendation

---
**Input** : User history $\mathcal{H}_u$, candidate set $C_u$, prompting scheme $q(\mathcal{P}_u|\mathcal{H}_u, C_u)$
**Output:** Total uncertainty, Recommendation uncertainty

1 Draw prompts $\{\mathcal{P}_u^n\}_{n=1}^5$ from $q(\mathcal{P}_u|\mathcal{H}_u, C_u)$
2 Generate logits $z_i^n$ with $\mathcal{P}_u^n$
3 Estimate $\hat{q}(\pi_u|\mathcal{P}_u^n)$ (Eq.5)

/* Total Uncertainty                    */
4 Estimate $\hat{q}(\pi_u|\mathcal{H}_u, C_u)$ with $\{\hat{q}(\pi_u|\mathcal{P}_u^n)\}_{n=1}^5$ (Eq.6)
5 Estimate total uncertainty with $\hat{q}(\pi_u|\mathcal{H}_u, C_u)$ (Eq.7)

/* Recommendation Uncertainty           */
6 Estimate conditional entropy $\mathbf{H}[\hat{q}(\pi_u|\mathcal{P}_u^n)]$
7 Estimate recommendation uncertainty with $\{\mathbf{H}[\hat{q}(\pi_u|\mathcal{P}_u^n)]\}_{n=1}^5$

---

We denote the second term as *recommendation uncertainty* originating from the recommendation difficulty associated with $\mathcal{H}_u$ and $C_u$. The recommendation uncertainty measures the average uncertainty in recommendation over potential prompts, thereby neutralizing the impact of prompting. Uncertainty decomposition allows us to distinguish whether the uncertainty arises especially from the prompting scheme or the recommendation difficulty. The system can acquire a thorough understanding of the volatility in LLMs' generation, and deploy various strategies for lower uncertainty and enhanced recommendation. The entire procedure of our uncertainty quantification framework is described in Algorithm 1.

# 6 PREDICTIVE UNCERTAINTY INDICATES RELIABILITY OF RECOMMENDATION

## 6.1 Experiment Setup

We briefly summarize the experiment setup due to limited space. Please refer to **Appendix A for the experimental details.**

**Evaluation metrics.** It is noted that the uncertainty quantification does not affect recommendation performance. Instead, our goal is to yield effective uncertainty measures that should indicate the reliability of recommendations. We adopt Kendall's $\tau$ ($\tau$@$K$) [59] and Concordance Index (C@$K$) [65] to assess whether a recommendation with lower uncertainty yields higher NDCG@$K$ (N@$K$) than one with higher uncertainty.

**Dataset.** We use three real-world datasets, including MovieLens 1M[7], Amazon Grocery [49], and Steam [31]. These datasets have representative text (e.g., title, description) for each item and have been widely used for LLM-based recommendation [25]. For each user's history, we hold out the last item for testing and the penultimate item for validation.

**Base large language models.** We utilize three popular LLMs from different generations, including Llama3-8B [64], Gemma-7B [62], and Mistral-7B [29], which are the largest models that can be fine-tuned on our single NVIDIA A100-80G GPU. We additionally adopt GPT-3-Turbo [4] for zero-shot ranking.

**Implementation details.** We adopt following two scenarios:

---

[7]https://grouplens.org/datasets/movielens/

Table 1: Effectiveness of uncertainty measures estimated by ours and methods compared.

| Model | Method | MovieLens 1M | | | | | Amazon Grocery | | | | | Steam | | | | |
|---|---|---|---|---|---|---|---|---|---|---|---|---|---|---|---|---|
| | | $\tau$@5 | $\tau$@20 | C@5 | C@20 | N@20 | $\tau$@5 | $\tau$@20 | C@5 | C@20 | N@20 | $\tau$@5 | $\tau$@20 | C@5 | C@20 | N@20 |
| **Zero-Shot** Ranking on Randomly Retrieved Candidates | | | | | | | | | | | | | | | | |
| Llama3-8B | Label Prob. | 0.1276 | 0.1143 | 0.5561 | 0.5341 | | 0.1412 | 0.1326 | 0.6004 | 0.5654 | | 0.2312 | 0.2042 | 0.6517 | 0.6188 | |
| | Semantic Unc. | 0.1162 | 0.1072 | 0.5589 | 0.5387 | 0.5526 | 0.1624 | 0.1472 | 0.6137 | 0.5731 | 0.5134 | 0.2767 | 0.2381 | 0.6803 | 0.6328 | 0.6255 |
| | Verb. 1S top-1 | 0.1369 | 0.1261 | 0.5594 | 0.5474 | | 0.1677 | 0.1463 | 0.6207 | 0.5737 | | 0.2161 | 0.1842 | 0.6324 | 0.5991 | |
| | Ours | **0.1512** | **0.1378** | **0.5867** | **0.5737** | | **0.1913** | **0.1643** | **0.6286** | **0.5861** | | **0.3045** | **0.2811** | **0.6921** | **0.6591** | |
| Gemma-7B | Label Prob. | 0.0926 | 0.0686 | 0.5605 | 0.5354 | | 0.1498 | 0.1144 | 0.5995 | 0.5591 | | 0.2428 | 0.1992 | 0.6508 | 0.6042 | |
| | Semantic Unc. | 0.1046 | 0.0758 | 0.5684 | 0.5391 | 0.4703 | 0.1594 | 0.1212 | 0.6059 | 0.5626 | 0.4754 | 0.2609 | 0.2145 | 0.6621 | 0.6122 | 0.5500 |
| | Verb. 1S top-1 | 0.0834 | 0.0551 | 0.5471 | 0.5185 | | 0.1379 | 0.0926 | 0.5917 | 0.5501 | | 0.2393 | 0.1902 | 0.6471 | 0.5906 | |
| | Ours | **0.1524** | **0.1236** | **0.5945** | **0.5641** | | **0.2065** | **0.1701** | **0.6306** | **0.5885** | | **0.3135** | **0.2777** | **0.6918** | **0.6482** | |
| Mistral-7B | Label Prob. | 0.1004 | 0.0823 | 0.5606 | 0.5431 | | 0.0993 | 0.0781 | 0.5617 | 0.5405 | | 0.1202 | 0.1035 | 0.5733 | 0.5540 | |
| | Semantic Unc. | 0.1088 | 0.0896 | 0.5656 | 0.5469 | 0.5258 | 0.1146 | 0.0903 | 0.5712 | 0.5468 | 0.4884 | 0.1391 | 0.1197 | 0.5847 | 0.5624 | 0.5201 |
| | Verb. 1S top-1 | 0.1123 | 0.0932 | 0.5617 | 0.5433 | | 0.1023 | 0.0818 | 0.5674 | 0.5424 | | 0.1417 | 0.1208 | 0.5861 | 0.5635 | |
| | Ours | **0.1491** | **0.1368** | **0.5873** | **0.5720** | | **0.1648** | **0.1406** | **0.5999** | **0.5733** | | **0.1799** | **0.1646** | **0.6068** | **0.5865** | |
| GPT-3.5-Turbo | Label Prob. | 0.1353 | 0.1158 | 0.5687 | 0.5457 | | 0.2200 | 0.1762 | 0.6397 | 0.5914 | | 0.2106 | 0.1873 | 0.6261 | 0.5992 | |
| | Semantic Unc. | 0.1310 | 0.1162 | 0.5660 | 0.5459 | 0.5555 | 0.2255 | 0.1821 | 0.6431 | 0.5944 | 0.5150 | 0.2106 | 0.1892 | 0.6262 | 0.6002 | 0.5980 |
| | Verb. 1S top-1 | 0.1398 | 0.1227 | 0.5711 | 0.5592 | | 0.2176 | 0.1684 | 0.6367 | 0.5877 | | 0.2143 | 0.1907 | 0.6294 | 0.6068 | |
| | Ours | **0.1543** | **0.1472** | **0.5893** | **0.5782** | | **0.2570** | **0.2227** | **0.6568** | **0.6170** | | **0.2242** | **0.2101** | **0.6301** | **0.6138** | |
| **Fine-Tuned** Ranking on Candidate Generation Model | | | | | | | | | | | | | | | | |
| Llama3-8B | Label Prob. | 0.2537 | 0.2291 | 0.6341 | 0.6243 | | 0.3304 | 0.2981 | 0.6999 | 0.6612 | | 0.2708 | 0.2703 | 0.9541 | 0.9534 | |
| | Semantic Unc. | 0.2638 | 0.2391 | 0.6511 | 0.6336 | | 0.3267 | 0.2980 | 0.6933 | 0.6506 | | 0.2842 | 0.2827 | 0.9523 | 0.9518 | |
| | Verb. 1S top-1 | 0.2682 | 0.2491 | 0.6513 | 0.6362 | 0.5913 | 0.3286 | 0.2988 | 0.7035 | 0.6616 | 0.6562 | 0.2855 | 0.2834 | 0.9506 | 0.9504 | 0.9790 |
| | MC-Dropout | 0.2698 | 0.2478 | 0.6587 | 0.6356 | | 0.3316 | 0.2931 | 0.7068 | 0.6635 | | 0.2921 | 0.2848 | 0.9581 | 0.9558 | |
| | Ours | **0.2738** | **0.2639** | **0.6619** | **0.6423** | | 0.3634 | 0.3399 | 0.7189 | **0.6890** | | **0.2941** | **0.2934** | **0.9597** | **0.9590** | |
| | BNN* | 0.2723 | 0.2614 | 0.6617 | 0.6404 | 0.5901 | **0.3649** | **0.3443** | **0.7191** | 0.6883 | 0.6613 | 0.2924 | 0.2917 | 0.9586 | 0.9585 | 0.9813 |
| Gemma-7B | Label Prob. | 0.2057 | 0.1834 | 0.6207 | 0.5968 | | 0.2810 | 0.2489 | 0.6705 | 0.6323 | | 0.2702 | 0.2702 | 0.9545 | 0.9539 | |
| | Semantic Unc. | 0.2113 | 0.1900 | 0.6239 | 0.6003 | | 0.2903 | 0.2573 | 0.6761 | 0.6367 | | 0.2801 | 0.2800 | 0.9559 | 0.9552 | |
| | Verb. 1S top-1 | 0.2161 | 0.1907 | 0.6245 | 0.6075 | 0.5618 | 0.2746 | 0.2386 | 0.6617 | 0.6193 | 0.5905 | 0.2880 | 0.2885 | **0.9602** | **0.9590** | 0.9762 |
| | MC-Dropout | 0.2214 | 0.2069 | 0.6318 | 0.6094 | | 0.2807 | 0.2481 | 0.6696 | 0.6319 | | 0.2792 | 0.2780 | 0.9561 | 0.9558 | |
| | Ours | **0.2339** | **0.2183** | **0.6354** | **0.6166** | | **0.3277** | **0.2949** | **0.6965** | **0.6599** | | **0.2906** | **0.2904** | 0.9593 | 0.9588 | |
| | BNN* | 0.2198 | 0.2017 | 0.6302 | 0.6039 | 0.5623 | 0.3185 | 0.2892 | 0.6894 | 0.6557 | 0.5927 | 0.2897 | 0.2895 | 0.9579 | 0.9576 | 0.9775 |
| Mistral-7B | Label Prob. | 0.2404 | 0.2275 | 0.6352 | 0.6164 | | 0.3305 | 0.3031 | 0.6976 | 0.6655 | | 0.2591 | 0.2590 | 0.9494 | 0.9489 | |
| | Semantic Unc. | 0.2367 | 0.2236 | 0.6331 | 0.6142 | | 0.3412 | 0.3127 | 0.7049 | 0.6707 | | 0.2771 | 0.2770 | 0.9509 | 0.9504 | |
| | Verb. 1S top-1 | 0.2331 | 0.2200 | 0.6305 | 0.6073 | 0.5769 | 0.3247 | 0.2925 | 0.6891 | 0.6600 | 0.6471 | 0.2736 | 0.2727 | 0.9504 | 0.9501 | 0.9787 |
| | MC-Dropout | 0.2476 | 0.2307 | 0.6437 | 0.6224 | | 0.3448 | 0.3159 | 0.7077 | 0.6722 | | 0.2743 | 0.2735 | 0.9566 | 0.9554 | |
| | Ours | **0.2586** | **0.2494** | **0.6508** | **0.6331** | | **0.3764** | **0.3516** | **0.7255** | **0.6966** | | **0.2802** | **0.2801** | 0.9592 | 0.9588 | |
| | BNN* | 0.2566 | 0.2443 | 0.6502 | 0.6318 | 0.5782 | 0.3695 | 0.3461 | 0.7199 | 0.6931 | 0.6499 | 0.2794 | 0.2792 | **0.9598** | **0.9595** | 0.9810 |

*BNN utilizes an ensemble of *five* LLMs and thus is not a direct competitor of this work.

- **Zero-shot Ranking**: We use the pre-trained weights from Hugging Face. The candidates are randomly retrieved, following [25].
- **Fine-tuned Ranking**: We fine-tune LLMs with the low-rank adaptation [26]. The candidates are retrieved with BPR-MF [56].

For the inference, we set the maximum number of user history to 20 (MovieLens 1M, Steam) and 10 (Amazon Grocery). The number of candidates is set to 20 for all datasets, and the leave-one-out ground-truth item is guaranteed to be included unless we note it separately. We sample five prompts for each user, and the final ranking list is generated by sorting the average probability across the five prompts (i.e., $\mathbf{E}_{\mathcal{P}_u}[q(i|\mathcal{P}_u)]$ for $i \in C_u$).

**Methods compared.** We adopt various state-of-the-art and conventional methods to compute the total predictive uncertainty:

- **Label Prob. [20]**: It uses the probability associated with the predicted label (i.e., confidence).
- **Semantic Unc. [39]**: It then counts the proportion of answers to estimate the predictive distribution, which is utilized for the predictive uncertainty.
- **Verb. 1S top-1 [63]**: It constructs prompts for LLMs to explicitly produce their confidence (e.g., "Provide the probability that your answer is correct (0.0 to 1.0).").

- **MC-Dropout [16]**: It approximates BNNs with a single model and dropout. We generate five recommendations with dropout for each prompt.
- **BNN [12]**: It uses an ensemble of multiple models and conducts Bayesian inference for generation. In the experiment, we utilize an ensemble of *five* independently fine-tuned LLMs and thus it is not a direct competitor of this research line [39].

MC-Dropout and BNN are adopted only for the fine-tuned ranking task, as we cannot apply them on zero-shot LLMs.

**Inference overhead.** Our framework takes a nearly identical computational burden to Label Probability and Verb. 1S top-1, as the number of inferences is the same. Semantic Uncertainty, MC-Dropout, and BNN require five times more inferences than ours.

## 6.2 Effectiveness of Predictive Uncertainty

Table 1 shows the performance of total predictive uncertainty estimated by ours and the methods compared. We observe that the predictive uncertainty indeed indicates the reliability of recommendation generated by LLMs, with C@20 of over 0.95 in Steam. Ours and compared methods exhibit higher $\tau$@K and C@K for the fine-tuned ranking task than the zero-shot ranking task, as the LLMs are aligned with the ranking task via fine-tuning. Notably,

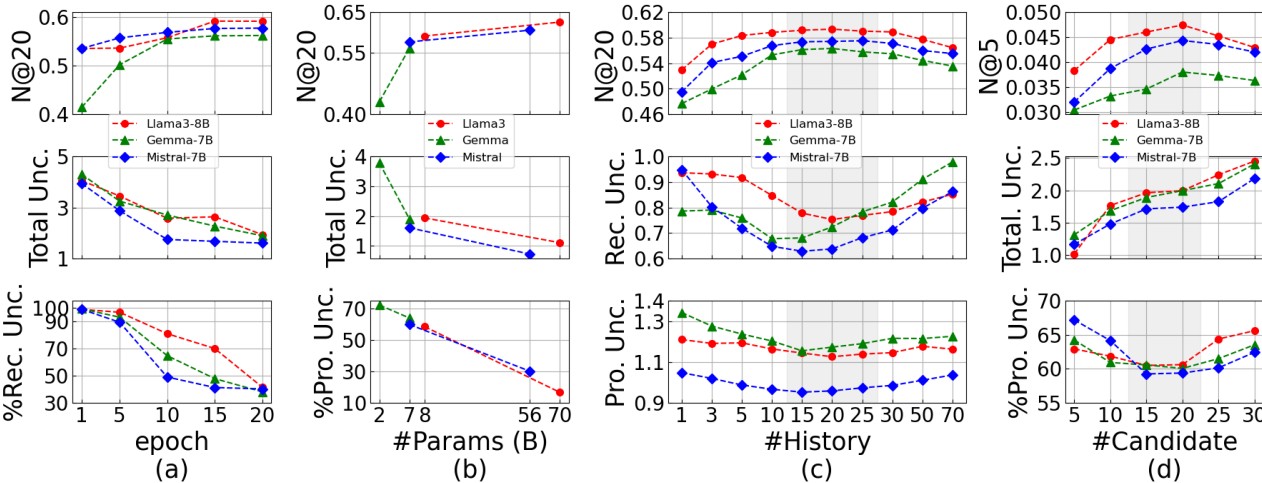

**Figure 4: Analysis of fine-tuned ranking on MovieLens 1M. %Rec. Unc. and %Pro. Unc. represent the proportion of recommendation and prompt uncertainty within the total uncertainty, respectively. For (d), the leave-one-out ground-truth item is *not* guaranteed to be included in the candidate set.**

our uncertainty measures outperform baselines in 18 out of 21 cells, demonstrating the superiority in indicating the reliability of LLM-based recommendations. Moreover, ours achieves comparable or even better performance compared to BNN, which utilizes an ensemble of five LLMs. In summary, the predictive uncertainty serves as an objective criterion for assessing the reliability of LLM-based recommendations, and can be further leveraged to develop various strategies aimed at enhancing user satisfaction.

## 7 UNVEILING ORIGINS OF UNCERTAINTY WITH UNCERTAINTY DECOMPOSITION

We further conduct thorough analyses for a deeper understanding on the origins of the uncertainty in LLM-based recommendations. Specifically, we investigate (a) epochs of fine-tuning, (b) the number of model parameters, (c) the number of user histories in the prompt, and (d) the number of candidates in the prompt. We explore how these factors affect each of the decomposed uncertainty measures and provide promising insights for the best use of LLMs' comprehension ability. The analyses are conducted in the fine-tuned ranking scenario on MovieLens 1M and implementation details are kept as done in Section 6.1. Figure 4 shows the recommendation performance and decomposed uncertainty for variations in generation and we emphasize four key observations as follows.

**O1: Total uncertainty decreases as the model is aligned with recommendation.** Figure 4a shows results for every five epochs of fine-tuning. As the model is fine-tuned to follow our instruction, the model gets aligned with the recommendation task [50], and therefore, the recommendation performance (N@20) is increasing. Since the total uncertainty indicates the reliability of the recommendation, it decreases as N@20 increases. Moreover, as the model is aligned with recommendation tasks, the proportion of recommendation uncertainty within the total uncertainty decreases as epochs progress. Lastly, we observe that predictive uncertainty is a

relative measure within the same model family, as Llama3-8B exhibits higher total uncertainty despite achieving a superior N@20 compared to the other models.

**O2: Larger models are more robust to the prompting scheme.** Figure 4b shows results for models with various numbers of parameters. We additionally adopt LLMs with various sizes, including Llama3-70B, Gemma-2B, and Mixtral-8x7B, and fine-tune these models on MovieLens 1M with LoRA. We observe that the recommendation performance (N@20) increases with the model size, as models with more training parameters have a higher ability to adapt to the given task [26, 44]. Accordingly, the total predictive uncertainty decreases, indicating the higher reliability of the generated recommendations. Moreover, the proportion of prompt uncertainty within the total uncertainty also decreases with the model size, demonstrating the larger models are more robust to the prompting scheme as observed in text classification [17].

**O3: Number of user histories affects recommendation uncertainty.** Figure 4c shows results for various numbers of user histories in prompts. It is obvious that more historical items can offer more information about the user preference, and hence, result in increased recommendation performance. However, the recommendation performance decreases with the user history larger than 25 items. This result is consistent with existing work on zero-shot ranking [23, 25] which demonstrates LLMs have difficulty understanding a long user history [78]. A large volume of user histories with congested tastes may complicate the assessment of user preferences, resulting in increased recommendation uncertainty. Our analysis stands as evidence that this phenomenon also occurs in fine-tuned LLMs, and our uncertainty decomposition framework offers a systematic way to gauge this adverse effect. We observe that LLMs tend to achieve higher recommendation performance when the number of user histories is associated with lower recommendation uncertainty. On the other hand, the prompt uncertainty remains almost unchanged, indicating that adjusting the number of chronologically ordered user histories has a marginal impact on it.

**O4: Number of retrieved candidates affects the proportion of prompt uncertainty.** Figure 4d shows results for various numbers of candidate items in prompts. We investigate the proportion of prompt uncertainty within the total uncertainty, since the absolute value of entropy increases with the dimension of output space (i.e., the number of candidates). In this analysis, the leave-one-out ground-truth item is *not* guaranteed to be included in the candidate set. As the number of candidates increases, the ground-truth item is more likely to be included in the candidate set, and thus the recommendation performance increases. On the other hand, we observe the recommendation performance decreases as the number of candidates gets larger than 20, which aligns with previous findings [55]. We provide a promising rationale for this issue from the perspective of prompt uncertainty; the number of possible permutations for candidate positions in prompts grows exponentially, and accordingly, the prompt uncertainty rapidly increases with large candidate size [15]. This increased prompt uncertainty cancels out the benefits of a larger candidate size (i.e., more likely to include the target item). We observe that the proportion of prompt uncertainty may serve as a barometer for balancing the trade-off between the benefits and detriments of increasing the number of candidates.

**Remarks.** Our analyses provide in-depth insights to support the best use of LLMs for recommendation. Specifically, we demonstrate adopting fine-tuning (O1) and larger models (O2) are beneficial to achieve lower predictive uncertainty and enhanced recommendation. Furthermore, our findings offer a promising explanation from the perspective of uncertainty; a larger number of user histories (O3) and candidate items (O4) may bring increased uncertainty and thus does not necessarily improve recommendation performance. In the following section, we demonstrate that the system can further enhance recommendations by constructing personalized prompts based on predictive uncertainty.

## 8 ENHANCING RECOMMENDATION WITH UNCERTAINTY-AWARE PROMPTING

Based on the previous analyses, we propose *personalized prompts* that adapt the number of histories and candidates to reduce the predictive uncertainty, thereby leading to enhanced recommendations. Due to a lack of space, we present experimental evidence on MovieLens 1M dataset. Results for Amazon Grocery and Steam datasets are present in Table 5, Appendix B.

**P1: Adjust the number of user histories in prompt based on recommendation uncertainty.** From **O3** in Section 7, we observe that the number of user histories affects the recommendation uncertainty. We construct prompts with varying numbers of user histories from {10, 15, 20, 25, 30}. Then, for each user, we choose the number of user histories that yields the minimum recommendation uncertainty. Table 2 shows the effectiveness of the uncertainty-aware user history adjusting for users with the top-5% recommendation uncertainty. The total/recommendation uncertainty decreases and the prompt uncertainty remains almost unchanged as shown in Figure 4c. Moreover, we emphasize that the average number of user histories used in prompts shows negligible change, as the selected number of user histories may decrease or increase from the default number.

**Table 2: Enhanced recommendation with uncertainty-aware prompting for fine-tuned LLMs on MovieLens 1M. #avg. denotes the average number of histories and candidates used.**

| Model | Prompt | N@20 | TU | RU | PU | #avg. |
|---|---|---|---|---|---|---|
| Uncertainty-aware **User History** Adjusting | | | | | | |
| Llama3-8B | default | 0.4231 | 2.485 | 2.016 | 0.469 | 20 |
| | unc.-aware | **0.4965** | **1.945** | **1.498** | **0.447** | 20.23 |
| Gemma-7B | default | 0.3998 | 3.041 | 1.570 | 1.471 | 20 |
| | unc.-aware | **0.4717** | **2.034** | **0.837** | **1.196** | 20.86 |
| Mistrial-7B | default | 0.4001 | 2.914 | 1.568 | **1.347** | 20 |
| | unc.-aware | **0.4843** | **1.956** | **0.573** | 1.383 | 19.41 |
| Uncertainty-aware **Candidate Set** Adjusting | | | | | | |
| Llama3-8B | default | 0.0658 | 2.371 | **0.853** | 64.01% | 20 |
| | unc.-aware | **0.0704** | **1.694** | 0.929 | **45.17%** | 20.46 |
| Gemma-7B | default | 0.0574 | 2.175 | 0.854 | 60.76% | 20 |
| | unc.-aware | **0.0671** | **1.489** | **0.789** | **47.01%** | 19.15 |
| Mistrial-7B | default | 0.0612 | 1.989 | 0.790 | 60.27% | 20 |
| | unc.-aware | **0.0669** | **1.300** | **0.751** | **42.22%** | 19.78 |

*For candidate set adjusting, PU represents the proportion within TU. Results for other datasets are present in Table 5, Appendix B.

**P2: Adjust the number of candidate items in prompts based on the proportion of prompt uncertainty.** From **O4** in Section 7, we observe that the number of candidates affects the proportion of prompt uncertainty. Similar to P1, we construct prompts with varying numbers of candidate items from {10, 15, 20, 25, 30}. It is noted that the leave-one-out ground-truth item is *not* guaranteed to be included in the candidate set for this analysis. Then, for each user, we choose the number of candidate items that yields the minimum proportion of prompt uncertainty. Table 2 shows the effectiveness of the uncertainty-aware candidate set adjusting for users with the top-5% prompt uncertainty proportion. We observe that uncertainty-aware candidate set adjusting yields enhanced recommendations with a lower proportion of prompt uncertainty and a lower total uncertainty. We also observe that the average number of candidate items used in prompts shows negligible change.

## 9 CONCLUSION

We highlight the need to improve our understanding of the reliability of LLM-based recommendations, considering a distinct challenge with the generation volatility of LLMs. We embrace the concept of predictive uncertainty as a quantitative measure of reliability, and introduce a novel framework to estimate it with a single inference. The estimated predictive uncertainty indicates the trustworthiness of LLM-based recommendations and is divided into recommendation uncertainty and prompt uncertainty. Through our in-depth analysis, we reveal key insights into how prompting details affect predictive uncertainty and recommendation performance. We further provide compelling explanations for the limitations of LLMs regarding a larger number of user histories and candidate items. Based on these insights, we propose to enhance recommendation performance with uncertainty-aware prompting approaches. We anticipate our work will aid efforts in improving the interpretability and explainability of LLM-based recommendations.

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

# Appendix

## A EXPERIMENTAL DETAILS

Our source code and model weights are available at https://anonymous.4open.science/r/UNC_LLM_REC

**Evaluation metrics.** Literature on the uncertainty of classifiers [39, 63] adopts AUROC to measure the probability that a randomly chosen correct prediction has a lower uncertainty than a randomly chosen incorrect prediction. In this work, similarly, we adopt Kendall's $\tau$ ($\tau@K$) [59] and Concordance Index (C@$K$) [65] to assess whether a recommendation with lower uncertainty yields higher NDCG@$K$ (N@$K$) than one with higher uncertainty.

$$\tau@K = \frac{1}{|\mathcal{U}|(|\mathcal{U}|-1)} \sum_{(i,j)} \text{sgn}(-\text{Unc}_i + \text{Unc}_j) \cdot \text{sgn}(\text{N@}K_i - \text{N@}K_j),$$

$$\text{C@}K = \frac{1}{|\mathcal{U}|(|\mathcal{U}|-1)} \sum_{(i,j)} \mathbb{1}(-\text{Unc}_i + \text{Unc}_j) \cdot \mathbb{1}(\text{N@}K_i - \text{N@}K_j). \quad (10)$$

For a user $i$, $\text{Unc}_i$ and $\text{N@}K_i$ denotes the estimated predictive uncertainty and NDCG@$K$, respectively. $\text{sgn}(\cdot)$ and $\mathbb{1}(\cdot)$ denote the sign function and the indicator function, respectively.

**Dataset.** We adopt the 20-core setting for MovieLens 1M and Steam, and the 10-core setting for Amazon Grocery. For each user's history, we hold out the last item for testing and the penultimate item for validation. The remaining items are used to construct $\mathcal{H}_u$. Data statistics after the preprocessing are presented in Table 3.

**Table 3: Data statistics after the preprocessing.**

| Dataset | #Users | #Items | #Interactions | Sparsity |
|---|---|---|---|---|
| MovieLens 1M | 6,040 | 3,883 | 1,000,207 | 95.74% |
| Amazon Grocery | 21,027 | 18,857 | 358,602 | 99.91% |
| Steam | 38,503 | 6,267 | 1,722,038 | 99.29% |

**Prompts.** The prompting scheme $q(\mathcal{P}_u|\mathcal{H}_u, \mathcal{C}_u)$ can be designed readily upon the interface of real-world applications. In this paper, we adopt the prompt in Figure 1 and randomly assign the position for candidate items. We modify verbs and nouns for each dataset (e.g., "watch" → "play" and "movie" → "game" in Figure 1 for Steam). We sample five prompts for each user, randomly assigning positions to candidate items for each of the five prompts. For the motivating analysis in Section 4, we devise four additional prompts in Figure 5 by slightly modifying the default prompt in Figure 1.

**Fine-tuning of base LLMs.** For each dataset, we fine-tune LLMs with the low-rank adaptation (LoRA) [26] for 20 epochs (MovieLens 1M and Amazon Grocery) and 10 epochs (Steam). For the fine-tuning, the prompt described in Section 5.2.2 is adopted, since we do not have ground-truth complete ranking lists. The number of user histories is sampled from {10, 15, 20, 25, 30} for MovieLens 1M and Steam, {6, 8, 10, 12, 14} for Amazon Grocery. The number of candidate items is sampled from {10, 15, 20, 25, 30} for all datasets. The learning objective is the next token prediction for the ground-truth item's index (e.g.,"B"). We adopt AdamW optimizer [46] and the learning rate is set to 2e-5 and the weight decay is set to 1e-2. For LoRA, we configure $r = 16$, $\alpha = 16$, and set the dropout rate

(a) "Candidate [A]" → "Movie [A]"

(b) "Candidate [A]" → "Candidate [1]"

(c) "Candidate [A]" → "Candidate A"

(d) "'{item}'" → "{item}"

**Figure 5: Modified prompts for motivating analysis in Section 4. {item} denotes the representative texts of an item.**

to 5e-2. We adopt 4-bit quantization for larger models (Llama3-70B and Mixtral-8x7B). All experiments are conducted on a single NVIDIA A100-80G GPU.

**Methods compared.** We adopt various state-of-the-art and conventional methods to compute the total predictive uncertainty:

- **Label Probability [20]**: It uses the probability associated with the predicted label (i.e., confidence) $\mathbf{E}_{\mathcal{P}_u}[\hat{q}(\pi_u^*|\mathcal{P}_u)]$.
- **Semantic Uncertainty [39]**: It stochastically generates multiple answers with a single prompt and obtains $\tilde{q}(\pi_u|\mathcal{P}_u)$ by counting the proportion of answers. We compute $\mathbf{H}[\mathbf{E}_{\mathcal{P}_u}[\tilde{q}(\pi_u|\mathcal{P}_u)]]$ with five answers for each prompt (i.e., total 25 inferences).
- **Verb. 1S top-1 [63]**: It constructs prompts for models to explicitly produce their confidence (e.g., "Provide the probability that your answer is correct (0.0 to 1.0)."). We use an average confidence of five prompts as reversed uncertainty.

**Table 4: Average recommendation performance of each prompting scheme used in the motivating analysis on MovieLens 1M.**

| Prompting Scheme | | Llama3-8B | | Gemma-7B | | Mistral-7B | |
|---|---|---|---|---|---|---|---|
| | | N@10 | N@20 | N@10 | N@20 | N@10 | N@20 |
| Same Prompt | zero-shot | 0.2145 | 0.3448 | 0.2100 | 0.3415 | 0.1889 | 0.3298 |
| | fine-tuned | 0.3958 | 0.4814 | 0.4058 | 0.4731 | 0.4402 | 0.5028 |
| Varying History Size | zero-shot | 0.2268 | 0.3569 | 0.2132 | 0.3433 | 0.1847 | 0.3272 |
| | fine-tuned | 0.5072 | 0.5717 | 0.4951 | 0.5485 | 0.5005 | 0.5553 |
| Varying Candidate Order | zero-shot | 0.2213 | 0.3533 | 0.2104 | 0.3414 | 0.1832 | 0.3257 |
| | fine-tuned | 0.5140 | 0.5769 | 0.5011 | 0.5538 | 0.5058 | 0.5591 |
| Slight Edit | zero-shot | 0.2221 | 0.3521 | 0.2109 | 0.3404 | 0.1912 | 0.3289 |
| | fine-tuned | 0.4641 | 0.5425 | 0.4374 | 0.5066 | 0.4496 | 0.5194 |

**Table 5: Enhanced recommendation performance with uncertainty-aware prompting for fine-tuned LLMs. TU and RU represent total uncertainty and recommendation uncertainty, respectively. PU denotes prompt uncertainty for user history adjusting and the *proportion* of prompt uncertainty for candidate set adjusting. #avg. denotes the average number of user histories and candidate items used for prompts. The ground-truth item is *not guaranteed* to be included in the candidate set for candidate set adjusting.**

| Model | Prompt | MovieLens 1M | | | | | Amazon Grocery | | | | | Steam | | | | |
|---|---|---|---|---|---|---|---|---|---|---|---|---|---|---|---|---|
| | | N@20 | TU | RU | PU | #avg. | N@20 | TU | RU | PU | #avg. | N@20 | TU | RU | PU | #avg. |
| | | Uncertainty-aware **User History** Adjusting | | | | | | | | | | | | | | |
| Llama3-8B | default | 0.4231 | 2.485 | 2.016 | 0.469 | 20 | 0.3748 | 3.852 | 2.715 | **1.137** | 10 | 0.7202 | 1.804 | 1.250 | **0.554** | 20 |
| | unc.-aware | **0.4965** | **1.945** | **1.498** | **0.447** | 20.23 | **0.4582** | **2.378** | **1.173** | 1.205 | 9.97 | **0.7353** | **1.372** | **0.807** | 0.565 | 19.79 |
| Gemma-7B | default | 0.3998 | 3.041 | 1.570 | 1.471 | 20 | 0.4079 | 3.204 | 2.674 | 0.530 | 10 | 0.7274 | 1.553 | 1.119 | **0.434** | 20 |
| | unc.-aware | **0.4717** | **2.034** | **0.837** | **1.196** | 20.86 | **0.4341** | **1.798** | **1.314** | **0.484** | 9.81 | **0.7404** | **1.190** | **0.743** | 0.448 | 19.82 |
| Mistrial-7B | default | 0.4001 | 2.914 | 1.568 | **1.347** | 20 | 0.3965 | 3.968 | 3.207 | **0.760** | 10 | 0.7508 | 1.496 | 1.041 | 0.455 | 20 |
| | unc.-aware | **0.4843** | **1.956** | **0.573** | 1.383 | 19.41 | **0.4498** | **2.502** | **1.304** | 1.198 | 10.02 | **0.7605** | **0.981** | **0.597** | **0.384** | 19.91 |
| | | Uncertainty-aware **Candidate Set** Adjusting | | | | | | | | | | | | | | |
| Llama3-8B | default | 0.0658 | 2.371 | **0.853** | 64.01% | 20 | 0.0517 | 2.707 | **0.844** | 68.82% | 20 | 0.0606 | **1.845** | **0.886** | 51.98% | 20 |
| | unc.-aware | **0.0704** | **1.694** | 0.929 | **45.17%** | 20.46 | **0.0632** | **2.317** | 0.959 | **58.61%** | 20.01 | **0.0714** | 1.848 | 1.031 | **44.21%** | 19.61 |
| Gemma-7B | default | 0.0574 | 2.175 | 0.854 | 60.76% | 20 | 0.0506 | 3.091 | 1.470 | 52.45% | 20 | 0.0621 | **1.639** | **0.751** | 54.20% | 20 |
| | unc.-aware | **0.0671** | **1.489** | **0.789** | **47.01%** | 19.15 | **0.0602** | **2.726** | **1.344** | **50.70%** | 20.07 | **0.0724** | 1.656 | 0.898 | **45.77%** | 20.14 |
| Mistrial-7B | default | 0.0612 | 1.989 | 0.790 | 60.27% | 20 | 0.0559 | 2.502 | 1.641 | 34.42% | 20 | 0.0641 | 1.526 | **0.638** | 58.21% | 20 |
| | unc.-aware | **0.0669** | **1.300** | **0.751** | **42.22%** | 19.78 | **0.0670** | **1.900** | **1.592** | **16.19%** | 19.61 | **0.0762** | **1.525** | 0.769 | **49.59%** | 19.83 |

- **MC-Dropout [16]**: It approximates BNNs with a single model and dropout. We obtain $\tilde{q}(\pi_u|\mathcal{P}_u)$ with five dropouts for each prompt and compute $\mathbf{H}[\mathbf{E}_{\mathcal{P}_u}[\tilde{q}(\pi_u|\mathcal{P}_u)]]$ (i.e., total 25 inferences).
- **BNN [12]**: It uses an ensemble of multiple models and conducts Bayesian inference for generation. In the experiment, we utilize an ensemble of *five* independently fine-tuned LLMs and thus it is not a direct competitor of this research line [39].

## B   ADDITIONAL EXPERIMENTAL RESULT

**Motivating analysis (Section 4).** Table 4 shows the average recommendation performance ($\mathbf{E}_{u \in \mathcal{U}}[\mu_{\mathrm{ndcg}}]$) of each prompting scheme used in the motivating analysis.

**Uncertainty-aware personalized prompting (Section 8).** Table 5 shows the effectiveness of our uncertainty-aware prompting for users with the top-5% recommendation uncertainty (User History

Adjusting) and users with the top-5% prompt uncertainty proportion (Candidate Set Adjusting). For user history adjusting, we construct prompts with varying numbers of user histories: {10, 15, 20, 25, 30} for MovieLens 1M and Steam, {6, 8, 10, 12, 14} for Amazon Grocery. For candidate set adjusting, we construct prompts with varying numbers of candidate items from {10, 15, 20, 25, 30}. Our uncertainty-aware personalized prompting decreases the predictive uncertainty and increases the recommendation performance, with negligible changes in the average number of user histories and candidates.