# OpenReview forum: "Uncertainty Quantification and Decomposition for LLM-based Recommendation"
_ACM.org/TheWebConf/2025/Conference — WWW 2025 Poster_

### Official Review · Reviewer_KQyE · 2024-11-25

**Novelty:** 5
**Technical Quality:** 4

**Review:**

This work focuses on an important and timely research problem: how to assess uncertainty in LLM-based recommender systems. The paper first illustrates that the reliability of a recommender system is closely aligned to its predictive uncertainty, then introduces an entropy-based method for assessing uncertainty. Additionally, it proposes an uncertainty decomposition framework to analyze the sources of uncertainty.

### Strengths

- The motivation is sound, and the research problem is very important. Understanding and assessing the uncertainty of the LLM recommendation is useful in practice.
- The motivation for technique is both intuitive and reasonable.
- The formula derivation is well-organized and easy to understand.

### Weaknesses

- The approximation based on top-1 probabilities in equation 5 lacks the theoretical guarantee. Given the autoregressive nature of LLMs, where each recommendation depends on both the input and preceding recommendations, this approximation conflicts with the Markovian dependency structure inherent in the generation process.  I look forward to the author’s response to determine whether or not to improve the technique score.
- While the uncertainty decomposition is interesting, the work has not illustrated any potential application of the decomposition. For example, how do we compute the prompt uncertainty and recommendation uncertainty in practice?
- The lack of the key hyper-parameter for generation. The uncertainty of LLM generation highly depends on the setting of hyper-parameters, such as temperature, top_p, and so forth. Please claim the value and discuss it in the paper.
- Minor: the font size of Figures 3 and 4 is too large.

**Questions:**

1. Can the authors provide more explanations for the latent variable $P_u$?
2. Why MC-dropout and BNN cannot be applied on zero-shot LLMs, as mentioned in sec 6.1?

**Reviewer Confidence:**

3: The reviewer is confident but not certain that the evaluation is correct

**Scope:**

4: The work is relevant to the Web and to the track, and is of broad interest to the community

---

### Official Review · Reviewer_2vsU · 2024-11-30

**Novelty:** 3
**Technical Quality:** 3

**Review:**

- This paper study the challenge of uncertainty in large language model (LLM)-based recommendation systems. The authors propose a novel framework for quantifying predictive uncertainty, which helps assess the reliability of recommendations generated by LLMs. They introduce a method for decomposing uncertainty into recommendation and prompt uncertainty, enabling a deeper understanding of the sources of instability in LLM-generated recommendations.

- The paper demonstrates that lower uncertainty correlates with higher recommendation performance and proposes uncertainty-aware prompting techniques to optimize recommendations by adjusting user history and candidate set sizes. Through extensive experiments, the authors show that their framework effectively improves the reliability and performance of LLM-based recommendations.

**Questions:**

- The solution proposed by the author is quite simple. It is only to adjust the user's history in the prompt. This scheme is too simplistic.
- The LLM itself has a certain degree of uncertainty. It seems that the author has not analyzed the impact brought by the inherent uncertainty of the large model.
- What are the sampling strategy and temperature coefficient used by the author when using LLM for recommendation?
- The paper highlights the role of fine-tuning in reducing uncertainty. How significant is the improvement in uncertainty and recommendation performance when fine-tuning LLMs compared to using a pre-trained model in zero-shot settings?
- Which one is more effective between the author's optimized prompt strategy and fine-tuning LLM? Or can the two be combined?
- If a larger LLM is used (e.g., 72B ), will the uncertainty of recommendation be reduced?

**Reviewer Confidence:**

3: The reviewer is confident but not certain that the evaluation is correct

**Scope:**

4: The work is relevant to the Web and to the track, and is of broad interest to the community

---

### Official Review · Reviewer_QuqH · 2024-12-02

**Novelty:** 4
**Technical Quality:** 3

**Review:**

The authors propose a framework to quantify and decompose predictive uncertainty in LLM-based recommendations, offering insights into its sources and introducing uncertainty-aware prompting techniques to mitigate it.

Advantages:

1. Provides a systematic approach to understanding and mitigating uncertainty in LLM-based recommendation systems, bridging gaps in reliability and interpretability while offering practical tools for improvement.

2. Offers valuable insights into factors influencing uncertainty, such as model size, fine-tuning, user history, and candidate set size, which can guide practitioners in optimizing LLM-based recommendations.

Shortages:

1. Incremental Novelty: While the paper adapts uncertainty decomposition and quantification techniques to LLM-based recommendations, its originality is limited compared to prior foundational works in this domain. For instance, [1] pioneered the decomposition of uncertainty in Bayesian deep learning, separating predictive uncertainty into aleatoric (data-related) and epistemic (model-related) components. This paper decomposes the uncertainty into prompt uncertainty and recommendation uncertainty. The Plackett-Luce model and MCMC are used to estimate recommendation uncertainty similarly to [1].  For prompt uncertainty, [2] explored uncertainty quantification in context learning for large language models. By comparison, the current paper applies related concepts but offers only a practical extension to recommendation tasks rather than advancing the state of uncertainty quantification for LLMs.

2. Prompt Sampling: Computing mutual information I involves estimating conditional entropies H, which rely on accurate sampling of prompts. The quality of this estimation depends heavily on the diversity and representativeness of the sampled prompts and whether the sampling process accurately captures real-world variability in prompts. The paper assumes the prompts are sampled from a well-defined distribution. However, it does not explicitly define the distribution, leaving some ambiguity about the assumptions behind this sampling process. Further, the sampling algorithm employed in the paper appears heuristic and lacks a rigorous theoretical justification. Specifically, the authors mention sampling 5 prompts from the distribution to estimate the predictive uncertainty and decompose it into recommendation and prompt uncertainties. However, sampling exactly five prompts is not clearly explained or justified.

3.  Simplified Assumptions in Ranking Models: The Plackett-Luce model assumes that items are selected independently at each step of the ranking process, which might not align with real-world scenarios where dependencies between items exist (e.g., correlated preferences).

4. Complexity: The paper lacks a detailed analysis of the computational complexity of its sampling algorithm, leaving unclear how it scales with the size of user histories, candidate sets, and prompts.

5. Limited Exploration of Diverse Prompting Strategies: While the paper focuses on uncertainty-aware prompting, it could benefit from a broader exploration of advanced prompting techniques (e.g., few-shot or instruction tuning) to mitigate prompt uncertainty.

6. Paper Quality: the paper alternates between terms like "volatility," "uncertainty," and "entropy" without explicitly clarifying their relationships. The paper should improve grammar and reduce redundancy for better quality.

References:
[1] Depeweg, Stefan, et al. "Decomposition of uncertainty in Bayesian deep learning for efficient and risk-sensitive learning." International conference on machine learning. PMLR, 2018.

[2] Ling, Chen, et al. "Uncertainty Quantification for In-Context Learning of Large Language Models." Proceedings of the 2024 Conference of the North American Chapter of the Association for Computational Linguistics: Human Language Technologies (Volume 1: Long Papers). 2024.

**Questions:**

NA

**Reviewer Confidence:**

3: The reviewer is confident but not certain that the evaluation is correct

**Scope:**

4: The work is relevant to the Web and to the track, and is of broad interest to the community

---

### Official Review · Reviewer_Gfxo · 2024-12-02

**Novelty:** 4
**Technical Quality:** 4

**Review:**

+ Quality:
This paper addresses the important issue of uncertainty quantification in LLM-based recommendation systems. Specifically, it proposes a solution by decomposing the predicted uncertainty into recommendation uncertainty and prompt uncertainty to deeply analyze the reasons behind the uncertainty in LLM-based recommender systems. Furthermore, the authors suggest that reducing prompt uncertainty can further improve the performance of the recommendation model. The paper presents extensive experimental results to demonstrate the effectiveness of the proposed method. However, the current work only uses three relatively small-scale LLMs and does not extend to larger models to prove the generalizability of the conclusions. Additionally, the paper lacks a demonstration of the experimental costs associated with measuring the uncertainty in LLM recommendations, making it difficult to determine whether this method is practical for real-world recommendation scenarios.

+ Clarity:
In scenarios with a smaller number of parameters, foundational language models often exhibit a limited understanding of text prompts. This can result in unstable outputs that frequently fail to meet the desired requirements, including issues such as hallucinations and format errors. I believe the authors need to further discuss these issues.

+ Significance:
In my view, this method is difficult to prove as practical for real-world scenarios.

**Questions:**

+ Q1: In recommendation scenarios, continuous model updates are often required, necessitating a trade-off between computational cost and model performance. What is the computational cost of the LLM recommendation uncertainty measurement proposed in the paper? How is the generalizability of this method demonstrated?

+ Q2: There are different strategies for adapting LLMs into recommendation systems. For instance, in LC-Rec [1], item textual information is quantized before performing LLM SFT. How can it be proven that the uncertainty measurement remains effective in these methods?

[1] Zheng B, Hou Y, Lu H, et al. Adapting large language models by integrating collaborative semantics for recommendation[C]//2024 IEEE 40th International Conference on Data Engineering (ICDE). IEEE, 2024: 1435-1448.

**Reviewer Confidence:**

3: The reviewer is confident but not certain that the evaluation is correct

**Scope:**

3: The work is somewhat relevant to the Web and to the track, and is of narrow interest to a sub-community

---

### Official Review · Reviewer_LrA8 · 2024-12-02

**Novelty:** 5
**Technical Quality:** 5

**Review:**

The paper introduces a framework for Uncertainty Quantification and Decomposition in LLM-based Recommendations to ensure the reliability of recommendations generated by large language models (LLMs). It aims to address the volatility and unpredictability in LLM-based recommendation systems caused by varying prompts and user histories.

Strength
- The introduction of predictive uncertainty for LLM-based recommendations fills a critical gap in understanding the reliability of such systems.
- The authors offer practical strategies like uncertainty-aware prompting to enhance recommendations
- Extensive experiments on diverse datasets validate the proposed framework's effectiveness.

Weakness
- The difference of prompts is not significant. Given the history and candidates, even with a different prompt template, the LLM should be able to generate consistent recommendation. The authors should consider more significant variations.
- The study does not explore how user profiles impact uncertainty in recommendations.
- The computational cost of obtaining uncertainty-aware prompts is unclear.

**Questions:**

- During adjusting the number of candidate items in prompts, how to choose the number of candidate items? Is it required to calculate the proportion of prompt uncertainty of every options?

**Reviewer Confidence:**

3: The reviewer is confident but not certain that the evaluation is correct

**Scope:**

3: The work is somewhat relevant to the Web and to the track, and is of narrow interest to a sub-community